# Diet Quality among Students Attending an Australian University Is Compromised by Food Insecurity and Less Frequent Intake of Home Cooked Meals. A Cross-Sectional Survey Using the Validated Healthy Eating Index for Australian Adults (HEIFA-2013)

**DOI:** 10.3390/nu14214522

**Published:** 2022-10-27

**Authors:** Yumeng Shi, Amanda Grech, Margaret Allman-Farinelli

**Affiliations:** 1Charles Perkins Centre, The University of Sydney, Sydney, NSW 2006, Australia; 2School of Life and Environmental Sciences, Faculty of Science, The University of Sydney, Sydney, NSW 2006, Australia; 3Susan Wakil School of Nursing and Midwifery, Faculty of Medicine and Health, The University of Sydney, Sydney, NSW 2006, Australia

**Keywords:** diet quality, food insecurity, nutrition insecurity, young adults, nutrition, 24-h recall, HEIFA-2013, university, college

## Abstract

Poor diet quality is commonly reported in young adults. This study aimed to measure the diet quality of students attending a large Australian university (including domestic and international students), and to examine the effect of food security status and other key factors likely to impact their diet quality. Using the Automated Self-Administered 24-h recall Australian version, a cross-sectional survey collected dietary recalls from domestic and international students in one university in Sydney. Diet quality was assessed using the validated Healthy Eating Index for Australian Adults (HEIFA-2013) which gives a score out of 100. Food security status was measured by the 18-item Household Food Security Survey Module. Differences in the mean HEIFA-2013 scores by student characteristics were determined by analysis of covariance. A total of 141 students completed one dietary recall. The mean HEIFA-2013 score for students was low (mean 52.4, 95% CI 50.0–54.8). Food-insecure students had a poorer diet quality (mean 43.7, 95% CI 35.7–51.8) than their food-secure peers (mean 53.2, 95% CI 50.8–55.7, *p =* 0.027). The mean HEIFA-2013 score was similar in domestic (mean 52.5, 95% CI 49.9–55.2) and international students (mean 51.9, 95% CI 46.3–57.5, *p =* 0.845). Those reporting self-perceived excellent cooking skills and higher cooking frequency had better diet quality. Interventions to improve food and nutrition knowledge and skills and address food insecurity may help tertiary education students cook more frequently and achieve better diet quality.

## 1. Introduction

Many young adults start to develop their independent lifestyle after the commencement of tertiary studies, including the management of their own meals [1,2]. Unhealthy dietary habits have been frequently reported among students attending tertiary education institutions in many high-income countries [3,4,5]. Some common unhealthy dietary practices included breakfast skipping, low consumption of fruits, vegetables, and whole grains, and high consumption of fast-food, saturated fat, added sugar, and sodium [3,4,5]. Low adherence to the Mediterranean dietary pattern (a healthy way of eating) was often found in students in European universities [6,7].

The overall diet quality has been measured in tertiary education students by using different diet quality indices based on national dietary guidelines (e.g., Healthy Eating Index) or specific dietary patterns (e.g., Mediterranean Diet Quality Index) [6,8]. Both sociodemographic and environmental factors have been reported to be associated with the diet quality of tertiary education students, including gender, living alone or with parents, ethnic backgrounds, socioeconomic status, and campus food environments [5,9,10,11]. Compared with domestic peers, international students experienced extra transitions including dietary acculturation when they relocate to a new country [12]. However, whether these additional challenges contribute to any differences in the overall diet quality between international and local students remains unclear.

The poorer diet quality of tertiary education students was found to be correlated with other unhealthy lifestyle behaviours and appeared to be associated with negative health and academic outcomes [13,14,15,16,17]. Lower physical activity levels and more screen time were observed in students with poorer diet quality [9,10,14]. Some studies showed that diet quality was negatively associated with the risk of metabolic diseases (e.g., obesity) in this student population [13,14]. Moreover, associations between diet quality and mental health were also observed. Quehl et al. reported higher levels of depression in female students with lower diet quality scores [15]. Furthermore, a positive association between diet quality and academic performance (e.g., grade point average) was noted [16,17].

Experiencing food insecurity is not uncommon in tertiary education students in Australia and other developed countries [18,19,20]. Having sufficient nutritious and preferred foods is important for achieving food and nutrition security [21]. International students tended to be more vulnerable to food insecurity than domestic students [12]. In the total community of tertiary education students, poorer dietary outcomes, e.g., lower intakes of fruits and vegetables, and higher intakes of added sugar, have been previously reported in food-insecure students compared with food-secure students [22].

To date, limited studies of diet quality and outcomes have been investigated among tertiary education students and analyses are mostly as an undifferentiated group [8,17]. In an Australian context, only separate dietary components, such as the consumption of fruits, vegetables, and takeaway foods, have been previously investigated in food-secure and -insecure tertiary education students [20], but not yet the overall quality of diet. This study aims to measure the overall diet quality of students (both domestic and international students) from an Australian university, and to compare the differences in the diet quality by food security status, cooking, sociodemographic, health, and environmental factors.

## 2. Materials and Methods

### 2.1. Study Design

As part of a larger cross-sectional food security survey conducted in a large Australian university in Sydney, New South Wales (NSW), we recruited participants to take part in a dietary survey to assess their diet quality. The study was approved by the Institutional Human Ethics Review Board.

### 2.2. Participants

Eligible participants included current domestic and international students aged 18 to 30 years who were not completing or had completed a nutrition course. The larger food security survey (unpublished) recruited participants from among university students through convenience and snowball sampling between October 2021 and May 2022. Advertisement flyers were posted on campus and through online platforms, e.g., announcements through an online learning management system of the institution and social media. Participants who completed the survey were asked if they had interest to participate in a dietary recall sub-study, and they were able to provide their consent through a secure online survey platform, Research Electronic Data Capture (REDCap, Version 12.2.1).

Of 467 respondents who completed the large survey, 258 participants provided consent to participate in the dietary recall sub-study. A drawing of ten $50 supermarket gift vouchers was conducted among the participants who completed dietary recalls. The value of the vouchers was not displayed in advertising materials.

### 2.3. Data Collection

Participants were asked to complete two 24-h dietary recalls (24HR) via the Automated Self-Administered 24-h Dietary Assessment Tool Australian version (ASA24-Aus), which is an online platform for completing self-administered dietary recalls and food records [23]. To collect dietary recalls, the ASA24-Aus followed a multiple-pass approach to collect information on food and drink items in each meal or snack session, preparation methods, additional ingredients, portion size, and forgotten items. The Australian Food, Supplement and Nutrient Database (AUSNUT) 2011–13 is applied in this ASA24 version [24]. Individual login details to ASA24-Aus were sent to each participant via email. Instructions and assistance were provided for participants to complete 24HR on two non-consecutive days, a weekday and a weekend day. Reminder emails were sent to encourage more participants to complete dietary recalls. The data collection opened from October 2021 until July 2022. This was impacted by the COVID-19 pandemic as Australian borders were closed for international until December 2021.

Student characteristics (e.g., age, gender, current academic degree, cooking frequency, and self-rated physical and mental health) were collected in the larger food security survey. Food security status in the past 12 months was assessed by using the 18-item Household Food Security Survey Module (HFSSM) from the United States (U.S.) Department of Agriculture [25]. Students with dependent children answered all 18 questions in the HFSSM [25], while students without children answered the first 10 questions only. The levels of food security were categorised into four categories based on the sum of affirmative responses to the HFSSM: high (no affirmative responses), marginal (1–2), low (3–7 for students with children, 3–5 for students without children), and very low food security (8–18 for students with children, 6–10 for students without children) [25]. High and marginal food security were then combined as the food secure group, while low and very low food security were combined as the food insecure group [25].

WHO-5 Well-being Index, a five-item tool, was used to further assess mental health status and it has been used in tertiary education students [26,27]. The total raw score ranges from 0 to 25, and a score below 13 is indicating poor wellbeing and further testing for depression [27].

### 2.4. Diet Quality Index

The Healthy Eating Index for Australian Adults (HEIFA-2013) was used to assess the overall diet quality of domestic and international students from an Australian university. The HEIFA-2013 was developed based on the most recent Australian Dietary Guidelines (ADG) in 2013 to reflect compliance with this national guideline [28]. This tool has been validated in Australian young adults through dietary data collected by weighed food records, food frequency questionnaires, and 24HR [29,30].

The HEIFA-2013 contains 11 components and the total score ranges from 0 to 100. The following nine components have a value of 10 marks for each, including fruits (5 marks for total intake and 5 marks for the variety of consumption), vegetables (5 marks for total intake and 5 marks for the variety), grains (5 marks for total intake and 5 marks for wholegrains intake), total intake of dairy products, total intake of meat and alternatives, total intake of discretionary foods, fat intake (5 marks for saturated fatty acids and 5 marks for mono- and poly-unsaturated fatty acids), sodium intake, and the intake of added sugar; with another 5 marks for each of the water and alcohol intake.

The criteria for full marks of each component followed the recommendations for male and female adults aged from 19 to 50 years in the ADG [28], and the serve size followed the Australian Guide to Healthy Eating (AGHE) [31]. The serve size with the criteria for full and no marks for each component are shown in Table 1. Marks were given incrementally between full and no marks. Taking dairy intake as an example, 2.5 serves or more will be given a full 10 marks, and then 8 marks for 2.0–2.4 serves, 6 marks for 1.5–1.9 serves, 4 marks for 1.0–1.4 serves, 2 marks for 0.5–0.9 serves, and no marks if less than 0.5 serve was consumed. Further details of this tool have been published previously [30,32].

### 2.5. Data Analysis

To calculate the HEIFA-2013 [30], the first dietary recall of each participant was used. A single dietary recall is a valid assessment of group means [33]. Of participants who completed the first recall, a subset completed the second recall. As an additional analysis, for participants who completed two recalls, the average intakes of two days were calculated and HEIFA-2013 determined (see Figure 1 for details of single and two-day recalls). To calculate food group intakes, mixed dishes were disaggregated into separate ingredients based on the recipe file from AUSNUT 2011–13 [24]. The fat and sodium intakes were derived from the output analysis file from the ASA24-Aus. The intake of added sugar was not one of the variables in the ASA24-Aus output and was calculated by merging the file of reported food items from participants with the nutrient file from AUSNUT 2011–13 and summing total daily intake for each person [24]. The calculation of the HEIFA-2013 was conducted via SAS version 9.4 (SAS Institute, Cary, NC, USA).

For misreporting, the ratio of energy intake to estimated basal metabolic rate (EI: BMR) was calculated for each participant [34]. The Schofield equation was used for estimating BMR based on self-reported weight [35], and three participants were excluded from the analysis as they did not report weight. Another two participants were excluded from the analysis because they did not specify male or female, and the HEIFA-2013 was a sex-specific measure. The mean HEIFA-2013 scores were adjusted by the continuous variable EI: BMR to allow for differences in accuracy of self-report and estimation of the diet quality and interpretation of the association between the diet quality and student characteristics.

The adjusted mean HEIFA-2013 total scores and differences by student characteristics were determined by analysis of covariance. The Mann–Whitney U test was used to compare the differences in the scores of individual components of the HEIFA-2013 by food security status because the data was not normally distributed. Statistical analysis was performed by using IBM SPSS Statistics version 25.0 (IBM Corp, Armonk, NY, USA).

## 3. Results

A total of 146 participants completed the first recall, with similar response rates in domestic (31%) and international students (30%). Of these students, a subset (*n =* 97) completed the second recall. After exclusion, 141 students (115 domestic and 26 international) who completed a single recall were included for data analysis, and the subset of 93 students (71 domestic and 22 international) who completed two recalls were included for an additional analysis (see Figure 1). 

Our sample (*n =* 141) was predominantly female (81%) and undergraduate (74%) students. More than half of participants (59%) lived at the parental home, and 69% were employed. Nine percent of students in this sample were food insecure.

From the first recalls, the mean HEIFA-2013 score was 52.5 (95% CI 49.9–55.2) in domestic students and 51.9 (95% CI 46.3–57.5) in international students after adjusting for EI: BMR (*p =* 0.845). The mean HEIFA-2013 scores of domestic and international students are shown by sociodemographic characteristics in Table 2 and by diet and health-related characteristics in Table 3.

Among sociodemographic factors, only changes caused by the COVID-19 pandemic were found to be associated with diet quality in our sample. Domestic students who experienced difficulties in finding employment due to the pandemic had a lower score for diet quality than those without such experiences. International students who experienced changes in living arrangements (e.g., moved to shared accommodations) had a higher score for diet quality than those without changes.

Students with food insecurity (mean 43.7, 95% CI 35.7–51.8) were found to have a lower score for diet quality than food-secure students (mean 53.2, 95% CI 50.8–55.7, *p =* 0.027), but this association was significant in only domestic students due to the insufficient statistical power in international students. Domestic students with self-perceived excellent cooking skills and higher cooking frequency (4 days or more per week) had a higher score for diet quality than those with poorer skills and lower cooking frequency. Better diet quality was also associated with better self-reported physical health status in domestic students.

A subset of students completed two recalls (*n =* 93, see Figure 1), and the HEIFA-2013 score for each of these students was calculated based on their average intake of two days. The mean HEIFA-2013 score in domestic (mean 51.9, 95% CI 48.9–54.9) and international students (mean 53.9, 95% CI 48.4–59.3) remained low and similar (*p =* 0.536). The mean HEIFA-2013 scores of domestic and international students by student characteristics from average intakes can be seen in Appendix A.

The scores of individual components in HEIFA-2013 are compared between food-secure and -insecure students in Table 4. Students with food insecurity tended to consume less wholegrains (based on first recalls) and more added sugar (based on average intakes from two recalls) than food-secure students.

## 4. Discussion

Poor diet quality was found in both domestic and international students attending an Australian university. Food insecurity was associated with lower diet quality score in domestic students. Among international students the association could not be confirmed due to a small sample size but there appeared to be some association. Self-perceived excellent cooking skills and higher cooking frequency were found to be protective factors for better diet quality. Solutions to alleviate food insecurity and enhance food literacy among tertiary education students may facilitate the improvement of their diet quality.

The association of food insecurity with poorer dietary outcomes in university students has been previously reported by a systematic review, although the results were not consistently significant across studies [22]. The association between the overall diet quality and food security status was rarely measured using validated dietary assessment tools in this student group [22]. In our study, significantly lower overall diet quality and whole grains intake were found in food-insecure domestic students compared with food-secure ones, but this association could not be confirmed in international students due to the small sample. From previous studies investigating the strategies tertiary education students used in coping with food insecurity, food insecure students often had to choose less expensive and less healthy alternatives [36,37]. In Australia, the prices of healthy food options were generally higher than less healthy options, and this has also been reported for food outlets on campus [38,39]. Moreover, the cost of living in Sydney where the study was conducted, remained the highest in the country [40]. The prevalence of food insecurity in Australia appeared to increase during the COVID-19 pandemic [41]. Food insecure students may not have sufficient income or financial support to consistently afford healthy meals.

Considering the association between food insecurity and poorer diet quality in our sample, food assistance services may be helpful for them. Food assistance services may improve the diet quality of their recipients when more fresh and healthy options were provided, e.g., fruits, vegetables, and whole grains, but the improvement might be limited if the provided options were in low nutritional quality [42,43]. More programs were provided for international students in NSW and on campus during the COVID-19 pandemic [44]. From the hunger report by Foodbank Australia in 2021 [45], dietary needs of people receiving food hampers were not fully achieved, such as meeting special dietary requirements and the need for cultural foods. A small number of students in our sample used these services, and thus the impact of using food assistance services on the diet quality of food-insecure students could not be judged from this study.

The overall diet quality of domestic and international students was generally low in our sample, but no significant differences were found between the two groups. A large-scale survey conducted in another Australian university reported different eating behaviours between domestic and international students, i.e., higher intakes of fruits and vegetables in domestic students, and higher intakes of grains and soft drinks in international students [46]. Compared with 24HR collected from the Australian Bureau of Statistics (ABS) 2011–2012 National Nutrition and Physical Activity Survey, the diet quality score of our students was slightly higher than among young adults from the national data with a total HEIFA-2013 score of 41.6 [32]. This may result from the higher socioeconomic status and tertiary education level of these students compared with the general population of young adults [32,47]. Students with more interest in food and nutrition might have been more willing to participate in our study.

The association between food preparation and better diet quality in young adults has been previously reported by an Australian study for students completing a nutrition degree and a population-based longitudinal study in the U.S [48,49]. Similarly, the higher cooking frequency was associated with better diet quality in our domestic students, although the association was not significant in international students. The previous Australian study also reported that lower frequency of takeaway and convenience meal consumption was associated with better diet quality [49], but the association between the frequency of eating out and diet quality was not detected here. The frequency of cooking and eating out among students might be affected by their cooking skills, time availability, and the pandemic [50,51]. Most choices from the menus of popular online food delivery outlets in Australia and New Zealand were found to be nutritionally poor [52].

The findings here imply that improving food literacy among tertiary education students may increase their confidence and frequency of cooking to achieve better diet quality. Increasing nutrition knowledge only might not be enough for students to translate this into dietary practices [9]. Food literacy covers more aspects of the required abilities for healthy eating, e.g., skills in planning, management, selection, and preparation [53]. A four-week food literacy program was effective for Australian adults with low-to-middle income as participants were able to improve their food literacy and maintain their positive changes in dietary behaviours after the program [54,55]. Similar programs could be regularly organised by universities and tailored for tertiary education students with different characteristics to increase their abilities in preparing healthy meals when facing time and financial constraints.

Food environment and purchasing on campus may affect the diet quality of tertiary education students, although this was not measured in our study. Previous studies from the U.S. and Australia found more frequent purchases of foods and beverages on campus were associated with poorer diet quality and less healthy eating habits, such as less frequent breakfast and higher consumption of energy-dense, nutrient-poor choices [56,57,58]. Higher availability of healthy options with cheaper prices were often suggested by food environment audits on campus and student surveys [38,56,59].

The strengths of this study include the measurement of the overall diet quality among domestic and international students through 24HR and a validated assessment tool (HEIFA-2013), and the investigation of the association between food security status and diet quality among students in an Australian university. There are also several limitations. The sample size of international students was small, and this contributed to insufficient power to detect statistical significance between different characteristics in this student group. In comparison with the national data on tertiary education students in Australia in the academic year of 2020 [60], our sample had an overrepresentation of females (81% in our sample vs. 56% in Australia) and undergraduate students (74% vs. 66%), and an underrepresentation of international students (18% vs. 30%). The findings from this study might not be generalisable to university students in other areas of Australia. This is a snapshot survey and used one-day recalls from participants to estimate group means. For participants who completed two recalls, the average intake of two days was used in this study, while a different statistical method was used to estimate usual intake from national surveys with large sample sizes [61,62]. Causal statements cannot be made due to the nature of the cross-sectional design.

To improve the diet quality of tertiary education students, solutions to address food insecurity and increase cooking skills could be helpful. Tailored food literacy programs, including regular nutrition education and cooking classes, may help students learn and use nutrition knowledge and increase their abilities for cooking. More interventions to improve the availability of affordable healthy meals on campus might also be beneficial for students with limited time for cooking. Food assistance services may need to be more available for food-insecure students, and the nutritional quality and cultural appropriateness of provided foods may require improvements to meet different dietary needs of students. For future research, a larger sample of international students could be recruited. This study was conducted during the COVID-19 pandemic and less international students studied onshore in Australia due to the border closures. Future longitudinal studies could be conducted to compare what happens to the diet quality and food insecurity in tertiary education students after the global pandemic.

## 5. Conclusions

In conclusion, diet quality was generally poor in both domestic and international students. In domestic students, food insecurity was associated with poorer diet quality, and better cooking skills and higher cooking frequency were associated with better diet quality. More contributing factors in international students might be detected by further studies with a larger sample. More specific programs to improve food literacy and address food insecurity in tertiary education students may facilitate the improvement of their diet quality.

## Figures and Tables

**Figure 1 nutrients-14-04522-f001:**
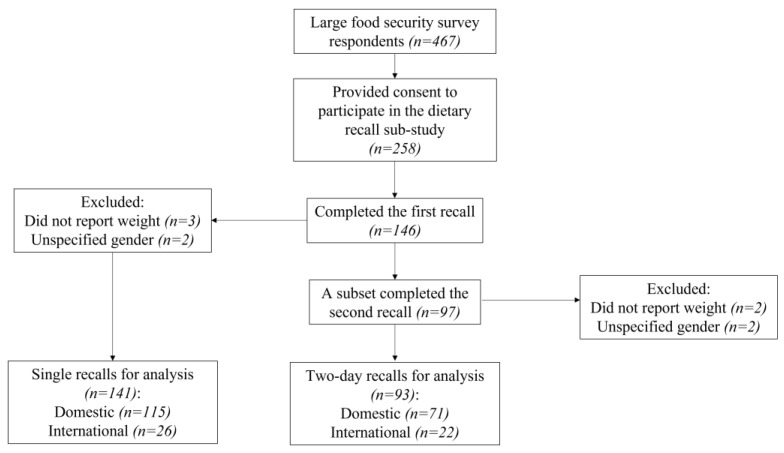
Flow diagram describing sample size.

**Table 1 nutrients-14-04522-t001:** Serve sizes and criteria for full and no marks of each component in the Healthy Eating Index for Australian Adults (HEIFA-2013).

Component	Serve Size/Unit	Criteria for Full Marks	Criteria for No Marks
Fruit			
Total intake	150 g fresh fruit, 30 g dried fruit, 125 g fruit juice	≥2.0 serves	<0.5 serve
Variety	150 g fresh fruit (pome, berry, citrus, stone, tropical, and other fruit)	≥2 types	<2 types
Vegetable			
Total intake	75 g vegetables and legumes, 125 g vegetable juice	Male: ≥6 serves, Female: ≥5 serves	<1 serve
Variety	75 g vegetables and legumes (green, starchy, orange, and other vegetable, and legumes)	≥1 serve of green, starchy, orange, and other vegetable, ≥0.5 serve of legumes (1 mark for each type)	<1 serve of each type (green, starchy, orange, and other vegetable) and <0.5 serve of legumes
Grains			
Total intake	500 kJ	≥6 serves	<1 serve
Wholegrains	500 kJ	≥3 serves	<1 serve
Meat and alternatives	65 g red meat/offal, 80 g poultry, 100 g fish/seafood, 120 g eggs, 170 g meat alternatives, 30 g nuts/seeds	Male: ≥3.0 serves, Female: ≥2.5 serves	Male: <1 serve, Female: <0.5 serve
Dairy	250 mL milk, 200 mL yoghurt, 550 kJ cheese, 200 mL custard	≥2.5 serves	<0.5 serve
Discretionary choices	375 mL beverages, 600 kJ foods	Male: <3.0 serves, Female: <2.5 serves	Male: ≥6.0 serves, Female: ≥5.5 serves
Fat			
Saturated	Percentage of energy intake	≤10%	>12%
Unsaturated	250 kJ	Male: ≥4.0 serves, Female: ≥2.0 serves	Male: <1.0 serves, Female: <0.5 serve
Sodium	mg	≤1610 mg	>2300 mg
Added sugar	Percentage of energy intake	≤5%	>10%
Water	Proportion of water consumed relative to other beverages	≥50%	<10%
Alcohol	1 standard drink (10 g of alcohol)	≤20 g of alcohol	>20 g of alcohol

**Table 2 nutrients-14-04522-t002:** Mean scores of Healthy Eating Index for Australian Adults (HEIFA-2013) by sociodemographic characteristics from domestic and international students attending a large Australian university ^1^.

Sociodemographic Characteristics	Total (*n =* 141)				Domestic (*n =* 115)				International (*n =* 26)			
	N (%)	Mean	95% CI	*p*	N (%)	Mean	95% CI	*p*	N (%)	Mean	95% CI	*p*
Age												
18–22 years	102 (72)	51.2	48.4–54.0	0.098	83 (72)	51.4	48.2–54.6	0.157	19 (73)	49.9	43.7–56.2	0.323
23–30 years	39 (28)	55.7	51.2–60.2		32 (28)	55.7	50.6–60.9		7 (27)	56.2	45.4–67.0	
Gender												
Female	114 (81)	52.3	49.6–54.9	0.779	93 (81)	52.3	49.3–55.4	0.679	21 (81)	52.0	46.1–57.9	0.770
Male	27 (19)	53.1	47.7–58.6		22 (19)	53.8	47.5–60.1		5 (19)	50.1	37.9-62.2	
Current academic degree												
Undergraduate	104 (74)	51.9	49.1-54.7	0.480	84 (73)	52.1	48.9-55.3	0.560	20 (77)	50.9	44.7-57.0	0.633
Postgraduate	37 (26)	53.9	49.2-58.5		31 (27)	53.9	48.7-59.2		6 (23)	54.1	42.4-65.8	
Marital status												
Never married	132 (94)	52.2	49.7-54.6	0.403	106 (92)	52.3	49.5-55.1	0.423	26 (100)	51.6	46.4-56.8	NA ^2^
Married or other	9 (6)	56.3	46.9-65.8		9 (8)	56.4	46.7-66.2		0 (0)	NA	NA	
Current accommodation												
Parental home	83 (59)	52.4	49.3-55.5	0.956	81 (70)	52.0	48.7-55.2	0.469	2 (8)	72.3	55.1-89.6	NP ^3^
Other than parental home	58 (41)	52.5	48.8–56.2		34 (30)	54.2	49.1–59.2		24 (92)	49.9	45.0–54.8	
Employment status												
No employment	44 (31)	52.8	48.5–57.1	0.829	28 (24)	53.2	47.6–58.7	0.823	16 (62)	52.1	45.3–58.9	0.821
Employed	97 (69)	52.3	49.4–55.1		87 (76)	52.4	49.3–55.6		10 (38)	50.9	42.3–59.5	
Weekly working hours												
Less than 20 h	77 (79)	51.3	48.1–54.6	0.200	69 (79)	51.7	48.2–55.1	0.343	8 (80)	49.2	35.5–62.8	NP
20 h or more	20 (21)	56.0	49.6–62.3		18 (21)	55.3	48.6–62.0		2 (20)	59.4	30.2–88.6	
Weekly income (AUD) ^4^												
$500 or less	98 (74)	51.1	48.2–53.9	0.053	82 (73)	51.1	47.9–54.2	0.071	16 (80)	50.8	43.1–58.5	NP
More than $500	35 (26)	56.6	51.8–61.4		31 (27)	56.7	51.5–61.9		4 (20)	57.8	40.3–75.3	
Changes caused by COVID-19												
Living arrangement ^5^												
No	123 (87)	51.6	49.1–54.2	0.083	104 (90)	52.3	49.4–55.1	0.431	19 (73)	48.2	42.7–53.8	**0.024**
Yes	18 (13)	57.9	51.3–64.5		11 (10)	56.0	47.1–64.8		7 (27)	60.8	51.6–70.0	
Job changes ^6^												
No	84 (60)	52.1	49.0–55.2	0.740	64 (56)	52.4	48.7–56.1	0.864	20 (77)	51.0	45.0–57.1	0.669
Yes	57 (40)	52.9	49.1–56.7		51 (44)	52.9	48.8–57.0		6 (23)	53.7	42.6–64.7	
Difficulties in finding jobs												
No	98 (70)	53.9	51.0–56.7	0.070	81 (70)	54.6	51.4–57.7	**0.028**	17 (65)	50.5	43.9–57.1	0.574
Yes	43 (30)	49.1	44.8–53.4		34 (30)	48.0	43.1–52.9		9 (35)	53.7	44.5–62.8	
Employment income loss												
No	87 (69)	51.7	48.6–54.8	0.524	69 (65)	52.0	48.5–55.6	0.702	18 (86)	50.3	43.3–57.4	NP
Yes	40 (31)	53.5	48.9–58.1		37 (35)	53.2	48.3–58.1		3 (14)	57.8	40.5–75.2	
Received any government support payment												
No	81 (59)	52.3	49.1–55.4	0.784	60 (54)	52.6	48.8–56.4	0.931	21 (81)	51.1	45.2–57.0	0.650
Yes	57 (41)	52.9	49.2–56.7		52 (46)	52.9	48.8–57.0		5 (19)	54.0	41.9–66.1	
Received any support payment from other sources												
No	130 (95)	52.0	49.5–54.5	0.352	108 (96)	52.3	49.4–55.1	0.222	22 (92)	50.8	44.9–56.6	NP
Yes	7 (5)	57.2	46.4–68.0		5 (4)	60.6	47.4–73.8		2 (8)	48.6	29.2–68.0	

^1^ Mean scores adjusted for energy intake (EI): basal metabolic rate (BMR) through analysis of covariance (ANCOVA), the potential range of HEIFA-2013 was 0–100; ^2^ Not applicable; ^3^ Not enough power; ^4^ Weekly income, include employment and/or other income; ^5^ Living arrangement changes, e.g., moved to less expensive premises; ^6^ Job changes, e.g., lost employment or worked less hours.

**Table 3 nutrients-14-04522-t003:** Mean scores of Healthy Eating Index for Australian Adults (HEIFA-2013) by diet and health-related characteristics from domestic and international students attending a large Australian university. ^1^.

Diet and Health-Related Characteristics	Total (*n =* 141)				Domestic (*n =* 115)				International (*n =* 26)			
	N (%)	Mean	95% CI	*p*	N (%)	Mean	95% CI	*p*	N (%)	Mean	95% CI	*p*
Food security status ^2^												
Food secure	129 (91)	53.2	50.8–55.7	**0.027**	105 (91)	53.5	50.7–56.3	**0.035**	24 (92)	52.1	46.6–57.6	NP ^3^
Food insecure	12 (9)	43.7	35.7–51.8		10 (9)	43.3	34.2–52.4		2 (8)	46.0	27.0–65.1	
Does your accommodation provide meals? ^4^												
No	48 (89)	52.2	48.5–55.9	0.662	26 (87)	54.6	49.1–60.1	NP	22 (92)	49.2	44.0–54.3	NP
Yes	6 (11)	49.7	39.2–60.3		4 (13)	46.6	31.9–61.2		2 (8)	57.6	40.5–74.7	
Weekly food budget (AUD)												
$0–30	18 (13)	48.2	41.6–54.9	0.157	18 (16)	48.3	41.5–55.1	0.129	0 (0)	NA ^5^	NA	0.610
$31–60	43 (30)	55.7	51.4–60.0		37 (32)	55.7	50.9–60.4		6 (23)	55.7	44.6–66.9	
$61–100	62 (44)	52.6	49.0–56.1		48 (42)	53.5	49.3–57.6		14 (54)	49.4	42.1–56.7	
>$100	18 (13)	48.4	41.8–55.0		12 (10)	46.2	37.8–54.5		6 (23)	52.7	41.6–63.8	
Adequacy of cooking facilities												
Adequate	125 (89)	52.6	50.0–55.1	0.727	107 (93)	52.9	50.1–55.8	0.398	18 (69)	50.6	44.3–57.0	0.559
Inadequate	16 (11)	51.2	44.1–58.3		8 (7)	48.3	38.0–58.7		8 (31)	53.9	44.4–63.4	
Self-perceived cooking skills												
Excellent	26 (18)	59.0	53.6–64.5	**0.015**	21 (18)	60.8	54.6–67.0	**0.009**	5 (19)	52.0	39.6-64.4	0.906
Good	62 (44)	49.4	45.9-52.9		48 (42)	49.0	44.9-53.1		14 (54)	50.6	43.2-58.1	
Fair or poor	53 (38)	52.7	48.9-56.5		46 (40)	52.6	48.5-56.8		7 (27)	53.4	42.8-64.0	
Cooking frequency												
3 days/week or less	75 (53)	49.1	45.9–52.2	**0.003**	65 (57)	49.3	45.8–52.9	**0.007**	10 (38)	47.2	39.0–55.4	0.170
4+ days/week	66 (47)	56.3	52.9–59.6		50 (43)	56.9	52.8–60.9		16 (62)	54.4	47.9–60.9	
Eating out frequency												
3 days/week or less	128 (91)	52.4	49.9–54.9	0.880	106 (92)	52.6	49.8–55.5	0.998	22 (85)	51.2	45.4–57.0	NP
4+ days/week	13 (9)	53.0	45.1–60.9		9 (8)	52.6	42.8–62.4		4 (15)	53.9	40.4–67.5	
Self-rated physical health status												
Excellent or good	85 (60)	54.6	51.6–57.7	0.027	67 (58)	55.2	51.6–58.7	0.031	18 (69)	52.6	46.1–59.1	0.593
Average or poorer	56 (40)	49.1	45.3–52.8		48 (42)	49.0	44.9–53.2		8 (31)	49.4	39.6–59.3	
BMI category ^6^												
Underweight	13 (9)	45.9	38.1–53.7	0.202	9 (8)	45.6	35.8–55.4	0.313	4 (15)	45.5	31.4–59.5	NP
Normal weight	101 (72)	52.7	49.9–55.6		81 (70)	52.9	49.6–56.1		20 (77)	52.3	46.2–58.4	
Overweight or obesity	27 (19)	54.4	48.8–59.9		25 (22)	54.4	48.3–60.4		2 (8)	57.2	38.0–76.4	
Self-rated mental health status												
Excellent or good	62 (44)	51.3	47.7–54.9	0.420	48 (42)	51.6	47.3–55.8	0.525	14 (54)	50.4	43.1–57.6	0.596
Average or poorer	79 (56)	53.3	50.1–56.5		67 (58)	53.4	49.8–56.9		12 (46)	53.1	45.3–60.9	
WHO-5 Well-being index category												
Normal	78 (55)	53.3	50.1–56.5	0.414	62 (54)	53.9	50.2–57.6	0.319	16 (62)	51.1	44.3–57.8	0.783
Poor wellbeing (below 13)	63 (45)	51.3	47.7–54.9		53 (46)	51.1	47.1–55.1		10 (38)	52.5	44.0–61.1	

^1^ Mean scores adjusted for energy intake (EI):basal metabolic rate (BMR) through analysis of covariance (ANCOVA), the potential range of HEIFA-2013 was 0–100; ^2^ Food security status was assessed by the 18-item Household Food Security Survey Module; ^3^ Not enough power; ^4^ A question for students lived outside parental or own home; ^5^ Not applicable; ^6^ BMI cut-offs, underweight (<18.5 kg/m^2^), normal weight (18.5–24.9 kg/m^2^), overweight (25.0–29.9 kg/m^2^) and obesity (≥30.0 kg/m^2^).

**Table 4 nutrients-14-04522-t004:** Median scores of individual components of Healthy Eating Index for Australian Adults (HEIFA-2013) by food security status among students attending a large Australian university.

HEIFA-2013 Individual Components	First Recalls ^1^			Two Recalls ^2^		
	Median (IQR ^3^)			Median (IQR)		
	Food Secure (*n =* 129)	Food Insecure (*n =* 12)	*p* ^4^	Food Secure (*n =* 86)	Food Insecure (*n =* 7)	*p* ^4^
Fruit total intake	0.0 (3.8)	0.6 (2.5)	0.667	1.3 (3.8)	1.3 (3.8)	0.926
Fruit variety	0.0 (5.0)	0.0 (0.0)	0.184	0.0 (5.0)	0.0 (5.0)	1.000
Vegetable total intake	2.0 (4.0)	1.5 (2.0)	0.620	2.0 (3.0)	2.0 (2.0)	0.349
Vegetable variety	1.0 (2.0)	1.0 (2.0)	0.454	1.0 (2.0)	1.0 (2.0)	0.808
Grains total intake	2.5 (2.9)	2.1 (3.1)	0.787	2.5 (1.7)	1.7 (2.5)	0.161
Wholegrains intake	0.0 (2.0)	0.0 (0.0)	**0.044**	0.0 (2.0)	0.0 (1.0)	0.648
Meat and alternatives	8.0 (7.0)	7.0 (10.0)	0.636	9.0 (6.0)	10.0 (8.0)	1.000
Dairy	4.0 (6.0)	2.0 (6.0)	0.141	4.0 (4.0)	2.0 (2.0)	0.421
Discretionary choices	7.5 (7.5)	3.8 (8.8)	0.228	7.5 (7.5)	5.0 (10.0)	0.559
Saturated fat	0.0 (5.0)	0.0 (4.4)	0.811	0.0 (2.5)	0.0 (2.5)	1.000
Unsaturated fat	5.0 (0.0)	5.0 (0.0)	0.725	5.0 (0.0)	5.0 (0.0)	0.383
Sodium	5.0 (10.0)	0.0 (9.0)	0.646	0.0 (5.0)	5.0 (10.0)	0.433
Added sugar	5.0 (8.0)	5.0 (10.0)	0.437	5.0 (10.0)	0.0 (5.0)	**0.006**
Water	5.0 (0.0)	5.0 (0.0)	0.739	5.0 (0.0)	5.0 (0.0)	1.000
Alcohol	5.0 (0.0)	5.0 (0.0)	1.000	5.0 (0.0)	5.0 (0.0)	0.631

^1^ Calculated the HEIFA-2013 based on the first dietary recalls of 141 students; ^2^ The subset of 93 students completed two dietary recalls, calculated the HEIFA-2013 based on the average intake of two recalls; ^3^ IQR = interquartile range; ^4^ Mann–Whitney U test.

## Data Availability

The data presented in this study are available on request from the corresponding author.

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
