# Peer review of "Diet Quality among Students Attending an Australian University Is Compromised by Food Insecurity and Less Frequent Intake of Home Cooked Meals. A Cross-Sectional Survey Using the Validated Healthy Eating Index for Australian Adults (HEIFA-2013)"

_nutrients, 2022, doi:10.3390/nu14214522_

Round 1

Reviewer 1 Report

Thank you for allowing me to review this manuscript investigating a study to measure the diet quality of domestic and international students attending a large Australian university, and to examine the effect of food security status and other key factors likely to impact their diet quality.

A brief summary – the manuscript was very well written, the ideas presented were clear and easy to follow.  The explanation of the methodology, results and conclusions all seem appropriate and relevant.  In addition, the authors were realistic about the limitations of their study and the conclusions drawn due to undertaking the study during a pandemic.

General concept comments

  • The flow diagram describing sample size was very detailed and easy to follow.
  • All tables were well organized and easy to understand.
  • It was helpful to see the comparison of outcomes with other studies.

Author Response

Thank you very much for reviewing our manuscript and providing comments.

Reviewer 2 Report

Introduction

This study provides a novel research question that will make a meaningful contribution to the area of college student health.

Methods

2.2 Participants

1. Are dietetics students the only nutrition-related majors at the universities sampled?

2. Suggest instead of “lucky draw” state “A drawing…”

3. The 18-item USDA FSSM was used – however, I am guessing some individuals were only asked the 10-item if no children were in the household, correct? It might be useful to clarify this, given not all students would have had children.

4. Better describe how response options are used to generate the four levels of food security and include the USDA citation with this. I would also recommend citing the USDA for collapsing the 4 levels into two (food secure and food insecure).

5. Recommend moving the sentences about the WHO-5 Well-being Index to a new paragraph.

2.5 Data Analysis

1. I am not understanding the first sentence – the HEIFA-2013 is a valid assessment of group means? Or one dietary recall is a valid assessment of group means? I think this sentence needs modified for clarity.

2. How many participants only completed 1 recall? How many two recalls? I see this information is presented in Figure 1. Suggest referencing that figure here.

3. The exclusions because of having sex-specific measures – is this because the respondents did not report their sex? I’m assuming yes from Figure 1, but it would be helpful to clarify this.

Results

Based on the study aim, I anticipated all analyses to compare domestic vs. international students. Instead, it appears that this was only done for HEIFA-2013 score. The remainder of the data is comparing variables – food insecurity, cooking skills, cooking frequency, self-reported physical health status, food consumption (e.g., added sugars, whole grains) – within domestic students and then within international students, but not comparing international to domestic students.

I am also confused why the authors then have a last analysis comparing food secure vs. food insecure students. The way the data are presented makes it hard to know if the authors were really interested in differences between international and domestic students or between food insecure and food secure students. However, I am guessing the analyses were not pursued further with comparing domestic vs. international because of the small sample size in the international sample. If this was the main study aim, it seems that the researchers would have aimed to achieve a higher sample of international students.

With such a small sample size of international students, perhaps the authors need to re-frame the purpose of this study to be around food insecure and food secure students (including both domestic and international students), and then offer rationale for the novelty of the analyses from other studies published in this area.

I also do not know what the second separate analysis with those who had two recalls added to the study. I am assuming the authors did not combine these data because it could add variability in having different numbers of recalls. If the HEIFA-2013 scores did not significantly differ between the two groups, perhaps that could be justification to analyze the data collectively.

 Discussion

The unique aspect of this study appears to be the diet quality measure evaluated between domestic and international students. However, because of the small sample size of international students, I am guessing this prevented any further analytical comparisons between domestic and international students. As mentioned earlier, I think the authors need to be more consistent with their study objective and data analyses performed.
